# Designing Digital COVID-19 Screening: Insights and Deliberations

**DOI:** 10.3390/ijerph20053899

**Published:** 2023-02-22

**Authors:** Soojeong Yoo, Natalia Gulbransen-Diaz, Callum Parker, Audrey P. Wang

**Affiliations:** 1Wellcome/EPSRC Centre for Interventional and Surgical Sciences (WEISS), University College London, London W1W 7TY, UK; 2School of Architecture, Planning and Design, The University of Sydney, Sydney, NSW 2006, Australia; 3Biomedical Informatics and Digital Health, School of Medical Sciences, The University of Sydney, Sydney, NSW 2006, Australia; 4DHI Laboratory, Research Education Network, Western Sydney Local Health District, Westmead Health Precinct, Westmead, NSW 2145, Australia

**Keywords:** COVID-19, health screening, hospital, health services, design, implementation, digital health, qualitative study, Internet of Things (IoT), Internet of Medical Things (IoMT)

## Abstract

Due to the global COVID-19 pandemic, public health control and screening measures have been introduced at healthcare facilities, including those housing our most vulnerable populations. These warning measures situated at hospital entrances are presently labour-intensive, requiring additional staff to conduct manual temperature checks and risk-assessment questionnaires of every individual entering the premises. To make this process more efficient, we present eGate, a digital COVID-19 health-screening smart Internet of Things system deployed at multiple entry points around a children’s hospital. This paper reports on design insights based on the experiences of concierge screening staff stationed alongside the eGate system. Our work contributes towards social–technical deliberations on how to improve design and deploy of digital health-screening systems in hospitals. It specifically outlines a series of design recommendations for future health screening interventions, key considerations relevant to digital screening control systems and their implementation, and the plausible effects on the staff who work alongside them.

## 1. Introduction

The COVID-19 pandemic highlighted the risk of emerging clusters of transmission in what is now an endemic and evolving disease [1,2] with new variants. Health systems internationally were severely challenged in their public health response to COVID-19 [3,4]. Discordance in infection prevention and control measures [5,6] existed whilst the world grappled with understanding the possible spread of COVID-19 symptoms. The pandemic has accelerated a plethora of digital health solutions through sheer necessity [7,8], allowing a natural experiment in adoption of technologies at a large scale, such as telehealth, enabling continuity of care that was safer for patients and staff [9,10]. Despite privacy concerns [11,12,13], many countries and jurisdictions used contact-tracing smartphone apps to keep track of the places an infected individual has been and their physical contact with other people [11,14,15]. These apps then usually alert an individual if a traced contact subsequently tests positive for COVID-19.

While contact tracing was one important strategy in the fight to keep track of COVID-19 infections, proactive screening for features such as COVID-19 symptoms might have provided an early signal of infection to alert people for further definitive testing, such as polymerase chain-reaction testing [16], and exclude them from entry [17]. This type of self-reporting is another measure that can be used for screening and has proven value compared to generalised digital syndromic surveillance; for example, when detecting respiratory symptoms and fever [18,19,20,21].

To aid with self-reporting, digital tools are emerging which collect self-reported symptoms to assist in reliably identifying risk factors associated with COVID-19 infection within a pandemic context [22,23,24]. Self-reported symptoms have been identified as potentially useful to collect within a digital health-screening system. Sudre et al. [25] collect their data via a self-reporting app and have provided a registry of emerging evidence in the frequency of reported symptoms for COVID-19 such as loss of taste and smell [26,27]. Self-reporting of symptoms could also assist in increasing self-awareness [28,29] of emerging threats and of the risk of coming into key high risk areas such as hospitals.

Pandemic control measures at hospitals include the utility of health screening for potential COVID-19 cases at entry points to limit and control who can come in. For example, a health care worker (HCW) such as a nurse may ask about symptoms and perform temperature checks on people entering the building. These manual screening processes are difficult to scale up as they can be labour intensive, risky, and inefficient in a time of scarce resources [30,31]. Therefore a practical, less human resource-intensive, non-contact solution to support our public health response is needed [32]. Despite this, the acceptance of digitally augmented public health-screening systems by HCWs that disrupt traditional clinical workflows remains underexplored. This includes how HCW acceptance affects authority and accountability when HCWs learn to use the technology and work alongside it.

To help address the gap, this paper reports on the deployment experiences of HCWs who took up the role of concierge screening staff stationed alongside a novel digital COVID-19 health-screening smart Internet of Things (IoT) system (hereafter: eGate) in a local children’s hospital. Through an analysis of qualitative data collected from interviews and questionnaires with 19 screening staff, we identify themes relating to the system’s design and deployment during the early formative evaluation phase crucial for identifying design implementation factors for the eGate [33,34]. The contribution of this work lies not only in these identified themes, but the considerations and recommendations for the design and deployment of future digital health-screening systems.

## 2. Augmenting the Health Screening Process

This research is framed around a proof-of-concept interdisciplinary project which aimed to digitally augment the manual COVID-19 health-screening process deployed at a local children’s hospital. This section provides background on the traditional (original) health-screening process used in the hospital during the very early stages of the COVID-19 pandemic (Figure 1 left). It then introduces the digital eGate screening process that uses Internet of Things technology [35], which was eventually deployed across the whole hospital (Figure 1 right).

### 2.1. Manual Screening Process

Before our project, groups of up to six concierge screening staff (hereafter: screening staff) were stationed outside each of the hospital’s entrances in order to screen and control access to the hospital. This manual screening process required the screening staff to physically screen staff and visitors face-to-face by asking them a series of risk assessment questions and taking their temperature (Figure 2 left). The questions were initially changed regularly based on updated state government’s health advice. The core questions asked, however, were whether the person being screened (screenee) had been to any of the recent COVID-19 case locations or exposed public transit route at a particular time, these were listed on sheets of paper posted on a wall and could, at times, span across multiple sheets (Figure 2 left). These questions were followed up with health-related questions specifically on whether the screenee was experiencing any flu-like symptoms based on the World Health Organisation’s (WHO) COVID-19 health advice (WHO COVID-19 Health Advice—https://www.who.int/emergencies/diseases/novel-coronavirus-2019/advice-for-public (accessed on 27 October 2020)).

Finally the screening staff would ask whether the screenee had been in contact with anyone who had tested positive for COVID-19 in the previous 14 days and whether they had been asked to be in isolation for the next 14 days. During this process, the screening staff would also take the screenee’s temperature with a handheld infrared forehead thermometer (temperature < 37.4 °C/99.32 °F to pass). If the individual was determined by screening staff as being healthy and passing the test, screening staff would give the person a coloured sticker (the colour changed daily) to affix on their clothes. The sticker was used to signify that an individual had passed the test for the day. This meant that they could leave the hospital and come back in that same day while avoiding the same screening process by showing screening staff their sticker.

The core challenges with this method, however, was that it was labour intensive, requiring manual temperature checks and risk-assessment questions. At a time where medical resources were already stretched, a practical and less human resource-intensive solution to support the public health response was needed, allowing more HCWs to return to the provision of direct patient care and minimise potential infectious exposure of staff and the public to COVID-19.

### 2.2. Egate Digital-Screening Process

Therefore, the goal of our project was to address challenges from the manual screening method through a digital health-screening system. This eGate system utilises a combination of evidence-based COVID-19 screening questions, temperature checks, and near real-time data analytics that can be updated as case locations and transport routes change based on daily government health updates. The eGate system’s deployment was formatively evaluated at one site entrance, before deployment occurred at all three entrances at the hospital. The system is composed of three key components: (1) a physical electronic gate; (2) temperature scanner; and (3) QR code scanner.

The system can be engaged with as follows (Figure 1 Right). Users need to answer all self-assessment questions on a smartphone app prior to hospital entry and save their QR token for the day. For example, this process might occur before they leave for the hospital to work or visit, such as whilst they are at home or whilst they are on the way to the hospital e.g., on a bus. This method puts the onus on the individual to assess their symptoms daily and check the daily COVID-19 case location updates prior to coming in. Once an individual arrives at the hospital, they need to present their QR token at the eGate. Following this, users will need to have their temperature checked by the eGate’s thermal scanner. Once these steps are complete, the eGate will make one of the following decisions:ALLOWED ENTRY: If temperature was below the pre-determined threshold e.g., less than 37.4 °C/99.32 °F, the physical gate opens to allow hospital entry. The profile behind the unique QR code is updated with temperature data and hospital pass entry is validated for the day. The QR code is printed on a physical sticker and collected by the participant as they enter the hospital. The printed sticker QR code is valid for one day and can be used as a way to gain faster entry through the gate without the need to go through the screening process again.PROHIBITED ENTRY: If temperature is out of the valid range e.g., more than 37.4 °C/99.32 °F, physical gate remains closed and the person is advised to see the screening staff. The profile behind the unique QR code is updated with temperature data and the hospital pass is defined as no entry for the day unless overridden by screening staff with an updated temperature reading from tympanic (ear-reading) temperature or overridden using the exception system by senior screening staff.

As shown in Figure 1 right, the eGate process does not require manual intervention. However, the hospital stationed one member of screening staff at each eGate that was deployed to ensure people were using the system properly and in case an individual screenee’s temperature was high in step 3.

## 3. Methods

At the time of writing, there were 69 healthcare workers in the local hospital who were currently working as screening staff. Screening staff included any available hospital staff with spare capacity during the initial lockdown measures including administrative staff, managers, and HCWs, such as dental assistants and nursing staff. Recruitment of screening staff through a roster system became the norm over time. This paper reports on the qualitative feedback we collected. This research was undertaken under ethical approval (HREC No: 2020/ETH02168) of the eGate system from the screening staff that had been working alongside it during technical tests, through interviews and questionnaires. This section first describes the eGate setup within the hospital, the data collection procedure, and then how the collected data was analysed.

### 3.1. Egate Setup

The eGate system was deployed initially at one site entrance (March 2021), then all three of the busiest public entrances to the hospital since May 2021. The eGate was specifically deployed at Entry Point 1, Entry Point 2, and Entry Point 3. Entry Point 1 and Entry Point 3 were more accessible to both staff and visitors, whereas Entry Point 2 is an entrance primarily used by staff due to its proximity to the staff carpark and offices. These entrances were major thoroughfares for the hospital, and to highlight the frequency of usage we have analysed the data for three months since deployment, 18 May–15 September 2021. We analysed three time periods based on the clinicians’ shift changing times: between morning (6 a.m.–12 p.m.), afternoon (12 p.m.–6 p.m.), and night (6 p.m.–6 a.m.).

The average usage for the eGate over the three months in the morning was 664 people (max: 1381 and min: 80). During the afternoon period (12 p.m.–6 p.m.) the average usage was 82 (max: 192 and min: 12). Night (6pm to 6am) had the least frequency of the eGate with an average of 55 people (max: 146 and min: 2). Monday had the most frequent usage (average: 664), while Saturday had the least (average: 638). eGate was the busiest during the morning time when health care workers arrive at their workplace and when visitors come for their doctor appointments.

### 3.2. Data Collection Procedure

Table 1 provides an overview of the participants and the data collection methods they were included in. We had 19 participants in total: 5 participated in the interviews and 14 in the online surveys. All the participants involved in these data collection methods had previous experience in the manual health screening process (before the eGate) in addition to their experience with being stationed alongside the eGate. Participants were all female and mostly young 18–25 (52.6%), 26–30 (15.8%), 31–40 (15.8%), and 40+ (15.8%). The screening staff had regular nursing roles outside of health screening, such as assistant, registered, and enrolled nurses. Data were collected over a period of 2 weeks (from Nov 2020). Advertisements for participants were sent out via internal emails and e-broadcasts, and snowball sampling recruitment methods used.

We now describe each of the data collection methods in detail.

#### 3.2.1. Interviews

Semi-structured interviews were utilised to further understand how five members of screening staff adapted to the deployed eGate system. Kvale [36] describes an interview as a “professional interaction” that provides insight into the world-perspective of the participant. This method offered particular value to this study given its incorporation of both open-ended and more theoretically driven questions, which allowed for us to explore pre-determined topics and any experiences reflexively raised by the participant [37]. The interviews themselves were conducted in situ with the screening staff working alongside the eGate at the time. Questions were oriented around the topics of usability (e.g., how difficult was the system to learn and understand?), user experience (e.g., how do you think staff/visitors perceive the gate?), and implementation (e.g., have you experienced any hardware/software failures? What did you do in that case?). General observations, raised by either the researchers or participants, were also discussed ad lib. The interviews lasted for 15 min, and the audio was recorded for later transcription and thematic analysis.

#### 3.2.2. Questionnaire

To develop a broader understanding of the eGate’s deployment, an online survey was distributed to screening staff who had experience working with the gate. Notably, this data was collected when local government health regulations restricted the researchers from being onsite themselves. This opt-in general survey allowed the research team to collect a larger series of responses than the semi-structured interviews (14 questionnaire respondents opposed to 5 semi-structured interviews), and more holistically describe the reality of the eGate’s deployment. Mills et al. [38] argues that researchers who want to know how their participants create and perceive their experiences will utilise open-ended statements and questions. To this end, the questionnaire was framed around understanding how the eGate screening compares to the manual screening process, how difficult it was to learn and understand, and identifying various problems experiences or improvements necessary to deliver a better experience.

### 3.3. Qualitative Data Analysis

The data collected from our interviews (transcribed by the researchers) and questionnaires were analysed following an inductive thematic analysis approach [39] to identify common themes. After an initial analysis, the themes were reviewed and cross-referenced by three researchers to iteratively refine and group them into higher level themes. The emerging themes allowed us to make sense of screening staff experiences with the eGate system and were used to structure our findings in the next section.

## 4. Findings and Recommendations

The thematic analysis of interview and questionnaire data resulted in two higher-order themes: (1) considerations for people; and (2) considerations for the system. Across these themes, we identify six key challenges and recommendations for overcoming them.

### 4.1. Considerations for People

Considerations for people span findings relevant to the shifting authority and accountability between screening staff and the eGate, learnability factors, means of troubleshooting and “hacking” the system, and accessibility factors identified by various individuals. Each theme is detailed further below.

#### 4.1.1. Shifting Authority and Accountability

The comments we collected from screening staff are reflective of a wider change in the dynamic between the staff supporting safe and screened entry into the hospital, and those attempting to gain access. In this section, we unpack findings relating to the shifting role of the screening staff due to the eGate, highlighting the change in authority and what it meant for accountability.

Before the eGate system was deployed, screening staff were responsible for manually screening individuals who wished to enter the hospital. However, the eGate’s deployment meant that the screening and decision-making process were primarily up to the system, unless a technical problem impaired its ability to make a decision on whether someone could pass through the entrance. Therefore, the role of screening staff has shifted to a support role for the eGate system where they needed to ensure people were using the eGate properly. However, due to the nature of the technology, the steps in the eGate’s screening process needed to be sequentially followed. As a result, one of the screening staff (P4) felt that the system took away their autonomy and the efficiencies that once came with it, “*before the eGate I could take people’s temperature [with a handheld temperature reader] while another screening staff conducted questionnaires, but now I can’t do anything even if they have the QR code ready as they need to be temperature checked by the gate*”.

While the eGate required accurate information to maintain safe entry conditions to the hospital, privacy was a frequently cited reason for user hesitation with the eGate. P4 mentioned that some users were “*unhappy to provide that info*” and felt that it was an “*invasion of privacy*”. Users of the eGate system were noted by P5 as being able to work around this issue altogether as it was “*easier for users to not be truthful*”. P5 further recounted an individual entering the hospital who intentionally misrepresented their health in the COVID eGate application. After registering no symptoms on their device, they were then asked how they felt by a screening staff working alongside the gate and replied “*I have been in bed with a cold*”. The participant then sought to confirm the individual’s health but was brushed off with the user arguing “*it was probably just allergies*”. On the other hand, participants recalled the manual screening process playing out differently, with P5 stating that “*we would’ve been able to observe that they had the symptoms*” and “*it’s harder for people to lie verbally [face to face] than what it is behind a screen*”. This is further argued, with one participant stating that a “*manual [COVID] hot spot check may force people to actually look through [the list] rather than skim through*”, shifting part of the onus from the nursing staff to the individuals entering the hospital. Similarly, another participant mentioned that the eGate “*[took] authority from nurses [screening staff] to make decisions*”, sanctioning the individuals entering the hospital as the one in control.

P5 reported that people would exploit vulnerabilities within the system “*angry parents walked through the crack [in the gates]*” and “*[when the gate is manually opened] some staff members push (sneak) their way through and we can’t stop them*”.

The authority the screening staff once had over the screening process was therefore lessened due to the eGate. One member of screening staff felt they had been relegated to technical support, which they felt fell outside their job descriptions: “*Look we don’t get paid to be engineers. We get paid to be nurses*” (P5).

While the authority appeared to have been shifted to the eGate for decision-making, screening staff were often held accountable for the system’s failures when users became frustrated. Notably, participants also recounted the verbal harassment they encountered when working alongside the eGate. Having transitioned the screening process from a team of multiple screening staff checking individuals at the door to a single member of screening staff working alongside each eGate at the three main entrances to the hospital, those individual screening staff quickly became a target to unhappy users. For instance, P2 mentioned that “*concierge staff get the blame when something goes wrong*” while P17 reported that they had experienced several negative interactions with other staff, “*rude behaviour, refusing to do the thermal temperature scans, refusing to list their last name (on the iPad), refusing to use their phone, verbally stating it is ridiculous to have to do this everyday just to name a few!*”. The abuse from frustrated users was reported to be particularly more frequent in the initial deployment of the eGate, described by P15 “*it was very hard for concierge staff in initial days as screening staff took the brunt of frustration from staff who weren’t aware of either the eGate in general or how to use it*”.

**Design Recommendation:*****Balance of power***Aligning the authority afforded to both the eGate and screening staff team can reduce the risk of total perceived authority being misconstrued to either party. Should the eGate be entirely undermined, pressure will return to the screening staff team to manually screen all individuals entering the hospital–an inconvenient and unsustainable system that positions multiple screening staff members on the metaphoric “front lines”. However, should all authority and power be given to *only* the eGate, individuals who find a way to “cheat” (be dishonest in their responses or strong-arm their way through the gate) could go unchecked. Therefore, ensuring that the screening staff team and eGate system work in tandem, presenting a united front of equal authority, is essential.  *This could look like:* making the physical presence of the screening staff team more imposing by positioning them as a component of the eGate rather than behind it, or the inclusion of a concealed button that directs users from the gate to the screening staff, allowing for the manual confirmation of a visitor’s screening.

#### 4.1.2. Learnability

All the screening staff agreed that the eGate system was “*effective and efficient*” (P12), easy for them to manage, with some describing it as “*not as difficult as it looked*” (P9) and “*common sense*” (P3). Two members of screening staff mentioned that they became confident with using the eGate system within two shifts. Despite there being no formal training with the eGate, according to P15 the screening staff were either “*self-taught or taught by others who also didn’t receive formal training.*” While this may not have posed a problem for those that worked permanently as screening staff to become adept with the eGate system as they had “*plenty of time and opportunity*”, P8 raised that staff working casually or after hours were not afforded the same opportunity to cultivate a sense of familiarity or predictability with the system, potentially leading to screening staff who “*may not know what to do if troubles come up*” with the system. However, due to this lack of formal training it was often mentioned that the eGate system was not always maintained properly, with tablets not being charged adequately from the previous screening staff, and that handovers between the screening staff were challenging.

When the eGate was initially deployed, staff who worked in the hospital were notified via email about how to use the eGate and could view instructional videos. However, P5 reported that not everyone received the email or just did not check them, and when “*asked to refer to it and they [staff] were upset*”. This resulted in screening staff “*taking the brunt of frustration from staff who were not aware of either the eGate in general or how to use it*” (P15). Since then, screening staff reported that users have mostly accepted the system after “*getting used to it*” (P1), even if it was somewhat reluctantly: “*staff would get cranky and not want to use it at first, but now use it even if they were upset with it*” (P4). Visitors (non-staff) however still reported to have issues with the system, not understanding the concept of putting their face into the frame of the temperature scanner or confusing the self-assessment app with the state government contact-tracing app (P16).

**Design Recommendation:*****Developing literacy through training***Despite initial attempts and distributing instructional material, this projects highlights the importance of planned and considered training initiatives alongside new technologies. Ensuring that training materials are consistently available to staff and visitors is a reasonable first step. However, careful consideration to the types of material offered is similarly worthwhile. As these resources serve to develop the digital literacy skills of a range of screening staff participants (e.g., casual staff who may not be exposed to the eGate frequently, visitors to the hospital who have limited digital and technological literacy skills) who will not only use the eGate themselves but also be responsible for supporting unfamiliar users.  *This could look like:* a dedicated series of training workshops courses or the provision of supplementary training material (i.e., videos and guides) which are accessible to staff offsite.

#### 4.1.3. Troubleshooting and “Hacking” the System

Screening staff who repeated experiences working with the gate quickly demonstrated a tenacity for troubleshooting and “*hacking*” the gate in order to ensure safe and screened entry into the hospital was facilitated, even in the face of either technical or interpersonal challenges. “*Hacking*”, in this instance, refers to the shortcuts and tricks utilised by the screening staff working alongside eGate when mechanical or network issues arose. These included manually keeping the gate open during network outages, activating the emergency open gate function during peak times (6 a.m. to 12 p.m.) to purportedly improve the flow, or covering the closing sensor in order to keep the gate temporarily open for longer e.g., a person had to cart goods through the gate. P17 provided further explanation around keeping the gate open during peak periods, explaining that it was an efficiency measure as the “*Gate took 10 s to open after a successful scan*”, despite the actual technical system process of five seconds or less.

If a user (visitor or staff member) arrived and had a problem with their phone (e.g., being an older model), the screening staff used a backup iPad to help the users screen their symptoms—“*I like the factor that we have a backup iPad so if someones phone does not scan or not work we can use [the] iPad*” (P1). While troubleshooting issues with the eGate was initially difficult, screening staff managed to overcome the difficulties “*once [they] completed that a few times [they] felt more confident*” (P14) and “*once you manage to fix one issue then you get confidence to use the system and fix it*” (P3).

To ensure troubleshooting knowledge and “hacking tips” were shared, a “*troubleshooting guide*” was co-created by the shift screening staff working with the gate and researchers. This included how to ask from help from senior screening staff to provide temporary QR code exceptions that would record the reasons for the exception. This was a living document updated and shared as necessary amongst the screening staff supporting the eGate in it is deployment. The participants’ often reported pragmatically solving their own problems encountered and once the fix was experiential learnt, the screening staff reported more confidence in their problem-solving abilities.

**Design Recommendation:*****Facilitate crowdsourced knowledge***The initiative demonstrated by screening staff to continuously adapt to and problem-solve with the eGate is an admirable reflection of the team. As such, the cultivation of these “hacks”, developed by those who worked the closest to the system is something that should be supported rather than stifled. In order to do this, facilitating the communication and dissemination of these “hacks” amongst screening staff and other staff is imperative.  *This could look like:* having a dedicated channel for communicating and sharing system management “hacks” or fostering a community within the hospital of dedicated “hackers”.

#### 4.1.4. Accessibility

The hospital is open to the public and the eGate was placed at its busiest entrances. This meant that people from all walks of life needed to therefore engage with it to gain entry. P1 and P4 recounted visitors from the older generation who struggled to adapt to this new system, particularly understanding the mental model of “*moving your [their] face into the square [frame]*” displayed on the screen of the temperature scanner. P1 speculated that manually using a hand-held infrared thermometer would be “*more familiar to people than the automated eGate scanner*”. Certain users were also reported to have experienced problems with scanning the QR code from their phone through the self-assessment app, with the screen automatically adjusting the interface orientation or turning off—“*not everyone knows how to use their phone*” (P2). This finding of not understanding the basics despite prolific smartphone ownership highlights the importance of having different pathways for less “tech savvy” people.

**Design Recommendation:*****Inclusivity***Inclusive design is generally important in the design and development of various interventions. However, given that the eGate is deployed in a hospital, whereby visitors with a greater diversity of abilities is common, these principles are particularly notable.  *This could look like:* developing and abiding by an accessibility guideline relevant to the hospital or increasing the diversity of participants in testing stages.

### 4.2. Considerations for the System

This section outlines findings relevant to eGate’s ergonomic and interactive affordances and the overall efficiency of the system. Detailed explanations are offered below.

#### 4.2.1. Ergonomics and Interactive Affordances

Furthermore, taller people were also reported by P1 and P2 as having issues getting their face into the frame, requiring manual adjustments to the camera by the screening staff. The consequence of people having issues understanding the technology or being unable to be detected by the eGate system led to inefficiencies, as summarised by P15—“*When people know how to use it, it works well. For those that don’t, it causes long lines and frustration*”. This finding highlights the importance of designing public systems that can adapt to different user characteristics [40].

Furthering this, screening staff commented on users (in this case, visitors and other staff members) struggling to operate and familiarise themselves with the correct operations of the system. In their questionnaire response, P15 outlined how the eGate was experiencing the “*usual issues faced when new technology is introduced–when people know how to use it, it works well. For those that don’t, it causes long lines and frustration*”. As a result of these challenges, inexperienced users quickly produced queues that subsequently extended entry time further. Other participants attributed these challenges to a “*clunky*” temperature-scanning device and a lack of knowledge with the system itself.

**Design Recommendation:*****Towards reactive systems***Similar to the recommendation previously detailed (*Inclusivity*), this recommendation focuses on ensuring the system can be utilised by the diverse range of visitors who frequent the hospital. By this, we mean that it is important for the system to not only effectively communicate how it should be used to its audience, but also respond to multiple forms of interaction in order to seamlessly guide an individual through its operations.  *This could look like:* integrating responsive feedback that makes individuals aware of and supports them in recovering from errors, or the facilitation of rapidly adaptable components that can be customised to the needs of each user.

#### 4.2.2. Efficiency

The intention for the eGate was to automate the screening process, bringing in efficiencies and allowing screening staff resources to be shifted to other important roles in the hospital. The consensus around the efficiency of the eGate system could be described by P3 as “*when it’s working, it’s very efficient*” and (P2) “*in a peak period it is really good*”. However, this efficiency came with the provision that everyone came prepared with their completed self-assessment questionnaires “*prior to lining up*” (P4) and had no issues using the system, brought on by lack of understanding or technical problems.

The eGate system itself was also reported to occasionally “lag” when processing an individual’s temperature, delaying the gate from opening. As a result of these inefficiencies, people needed to arrive to work earlier, with one participant reporting that they needed to come in 15 min early (P1). In order to help improve efficiency, participants suggested the eGate system should be scaled up during peak periods with the addition of a second eGate, more information for users on how to use the system, educating users to complete their questionnaire app independently before arriving, and staff only entrances—as staff are more familiar with the system and therefore can proceed through screening with fewer issues.

The eGate system was a research Internet of Things system prototype designed and developed by a small team at the University. As such, the team worked with the local hospital information and communications (ICT) teams to deploy contextual customisation and integration for the proof-of-concept system where allowed. However, the hospital has never had an Internet of Things system integrated into its network and communications infrastructure before. This meant that technical problems were experienced with the eGate system during iterative prototyping and initial deployment. These technical problems raised by the screening staff encompassed both hardware- and software-related problems, often resulting in periods of impeded screening efficiency. In terms of hardware, participants reported the eGate system’s physical gate not always opening as expected. When encountered, screening staff would need to reboot the entire eGate system, which reportedly took anywhere between 10 and 15 min. Along with the physical gate problem, the QR code scanning tablet did not always recognise an individual’s QR code and the tablet itself frequently needed changing, with one screening staff reporting they changed a tablet “*5 times during a 12 h shift*”. Participants also reported the printer jamming, requiring manual intervention. Furthermore, one participant mentioned that the printer would sometimes print misaligned QR codes on the stickers, having specifically “*about one in eight*” chance of occurring. Technical issues were particularly not received well by staff (users) when they “*came in all prepared and we say that the system is not working they get upset ‘oh this thing is not working again’*” (P4)

**Design recommendation:*****Redundancies***When critical eGate hardware or software was not functional, it was not always feasible for screening staff to address the problem at that time (e.g., during busy peak periods there is not enough time to try and reset the system lest they produce an even larger queue). In these scenarios, screening staff resort to a manual screening process without the eGate, instead writing down the staff or visitor’s details with pen and paper, and then providing them a coloured sticker (which alternated daily) to show they have passed screening for the day. This fallback was also used when a visitor arrived without a smartphone or had a smartphone that was not working properly. To preserve the function of the eGate, additional fallback measures should be integrated into the system, allowing for partial rather than full system failure.  *This could look like:* an additional set of controls which operate each feature in isolation or a partial-failure fallback whereby the system operates at partial capacity.

## 5. Discussion

### 5.1. Deliberations for Designing Screening Systems

Following the design recommendations, we outline five deliberations that are more broadly pertinent to designing health-screening systems.

#### 5.1.1. Prior Efforts and Preparation

During the design, development, and deployment of a health-screening system, it is firstly significant to maintain an awareness of the *prior efforts and preparation* expected of users. Health-screening systems, whether they are situated in a hospital, airport, or public space, provide entry to the venue. This could bottleneck the efficiency of the overall network if users are overwhelmed by the effort or preparation required by the eGate. To optimise ingress and maintain efficiency, designers should ask themselves: (1) what type of user labour is the system dependent on? (2) How easy is it to learn and fulfil this labour for the user? (3) What expectations of technical literacy might these expectations be contingent on?

#### 5.1.2. System Redundancies

Secondly, we champion the importance of *system redundancies* [41]. This refers to the idea that a health-screening system should be able to operate either fully or partially in any circumstance to maintain safe screening practices on the front line. Health-screening systems are implemented to support safe screening practices and, per their intention, actively reduce the number of screening staff who must be available at each entrance. However, consequently, should the eGate fail, the remaining screening staff would not be able to maintain the same effectiveness of operations without the gate, something greatly consequential to staff and visitors alike. Therefore, ensuring that the system is, firstly, adaptable to a diverse audience, and secondly, resilient to errors, is imperative. To action this, designers should consider: (1) how laborious it is to adapt components of the system to a range of users, (2) what technological redundancies are present in the system and what level of failure they account for, and (3) in the face of component failures, how support staff or users might access effective troubleshooting instructions and allow for continued use when technical experts may be unavailable.

#### 5.1.3. Visibility and Accuracy of Data

As previously described (Section 4.1.1), the eGate system shifts the authority and accountability of individuals involved. Despite this, the *visibility and accuracy of data* communicated between entrants of the hospital and the eGate/screening staff is of notable significance. Attention must be given to how users perceive and participate in the process of information sharing. In order for the eGate to be effective, the hospital is additionally reliant on the trustworthiness of an individual’s responses. However, this reliance must be negotiated with a person’s potential hesitancy to share personal data. Therefore, designers should consider: (1) to what extent is trust in the data provided necessary? (2) How will the tension between data accuracy and personal privacy be negotiated? (3) How will data reliability be managed?

#### 5.1.4. Scalability and Sustainability

This deliberation refers to the concern around long-term *scalability and sustainability* of a health-screening system. Consistent with most interventions, unique challenges arise when trying to scale or sustain the eGate system. In part, this has already been detailed, as each of the previous sections will become more complex and necessary as a health-screening system increases in scale and expected duration. However, we also highlight that emergent errors or issues can become exponentially more prevalent as a project scales, and given the well-justified risk-adverse context, the margin for error is slimmer than usual. Therefore, we encourage any act of scaling to be done with caution, and should only proceed when a firm, sustainable solution has been trialed and approved.

#### 5.1.5. Lowering the Risk of Exposure

Finally, we also offer reflections on an emergent theme. We believe the discussions surrounding the eGate’s potential to reduce screening staff’s exposure risk to COVID-19 to be of particular prominence, and a theme that should remain at the forefront of any intervention seeking to augment any health-screening system.

While screening systems are implemented within hospitals to keep patients, staff, and visitors safe, a team of screening staff are required to man a metaphorical front, an act which carries a substantial risk of infection [42]. In the previous manual screening system, screening staff at the hospital were required to take a visitor or staff member’s temperature by placing a handheld infrared temperature scanner close to the entrant’s forehead. This did not allow the screening staff to maintain a safe distance from a potentially infectious person. One of the key benefits of the eGate system, referenced by the screening staff, was that the eGate helped them keep at a safer distance from visitors and staff being screened. From the interview and questionnaire data, 5 participants explicitly mentioned that they felt safer with the eGate compared with the traditional screening method. Illustrating this perspective, their comments are as follows:

“*[The eGate] helps with social distancing - I don’t need to get closer with the person to check the temperature and I just need to hand over the sticker so it that way it is very efficient*”(P1)

“*It is good because we have a less contact its quicker system as well*”(P2)

“*Good thing is we don’t need to get up close and touch them, our role is to just guide them through the system*”(P4)

“*The concierge staff doesn’t need to stay close to check visitor’s temperature - able to maintain social distancing.*”(P18)

“*I think it is better in regard to social distancing and keeping screening staff safer*”(P9)

We encourage any future work developing similar screening interventions to maintain a clear and consistent understanding of their intention: to protect as many individuals as possible from adverse outcomes.

### 5.2. Interrelationship of Considerations

Practitioners know that the successful deployment of any designed intervention requires the careful balancing of multiple diverse and complex factors. Relevant to this project and future health screening interventions is the balance required between considerations relevant to people and considerations relevant to the system [43].

The aggregate themes emerging from this research can be explored across two scales: (1) focus-driven, and (2) progress-driven. The first scale was exemplified in the previous section, whereby findings were categorised according to the central focus of the design team’s efforts, i.e., aligning considerations to the system itself or the individuals working alongside the system. The second scale, which also offers valuable insight, is the temporal progress-driven lens. Using considerations for people as an example, we suggest that these findings can be further delineated depending on the stage of development the project is in, spanning from initial design efforts to the intervention’s deployment and maintenance. While some findings, such as *troubleshooting and “hacking” the system* are only evident when an intervention has been deployed and a person’s authentic interactions with the system can be observed over time, a consideration like *shifting authority and accountability* can be investigated at both the beginning of a design process (when these phenomena are first understood) and when the intervention is deployed (to observe how the system realistically mediates these phenomena).

Similarly, we recognise the dynamic nature of the focus-driven scale and revisit our findings. Exemplifying the interrelationship between considerations for people and considerations for the system are the findings related to ergonomics and accessibility. Comments related to the ergonomics of the system prioritised considerations of the system in a system/person dynamic, outlining (often physical) aspects of the system that resulted in negative experiences (screening staff noting that the temperature check screen was difficult to manually adjust for people of varying heights). On the other hand, comments referring to the accessibility of the system prioritised the perspective of people in the system–people dichotomy (screening staff recounting how older individuals struggled to position their face within the correct frame of the temperature check screen device). Nevertheless, the idea of the system–person relationship is present in both findings with any form of interaction between the system and people producing an impact on the other. In a less closely-knit example, we can also note that interrelationship between *troubleshooting and "hacking" the system* and *efficiency*, whereby the need to troubleshoot and “hack” the system is a direct consequence of efficiency errors.

Understanding the dynamic nature of the focus-driven and progress-driven scales is pivotal to the effective design and deployment of any intervention. However, before we can start implementing any recommendations relevant to each scale, or a specific finding, we also outline an integral understanding that emerged from this research: as dynamic as each focus-driven or progress-driven scale is, an additional relationship exists *between* the scales.

By this we mean that a combination of the scales would result in the following: system design considerations, system implementation considerations, design considerations influencing people, and implementation considerations affecting people. Again, as it is dynamic in the scale, so is dynamic *between* the scales. Evidencing this, we detail the influence of implementation considerations affecting people may have on system design considerations. An awareness of the long-term training capabilities of the stakeholders and their general capacity to communicate detailed and accessible training to all staff and visitors introduced to the eGate could heavily inform the extent to which ergonomic considerations inform the product design process. For example, knowing that a hospital may be under-resourced could encourage the prioritisation of a safe, usable, and accessible product, given that there is no formal opportunity to teach individuals about nuanced issues and their resolutions.

Finally, although the tension between the considerations of people and the considerations of a system are known, this research highlights that mastering the balance within and between the focus-driven and progress-driven scales is a far more complex and challenging activity.

### 5.3. Limitations

This research predominantly explores the perspective of screening staff (nurses and staff within the hospital). However, due attention should be given to the experiences of people being regularly screened, such as other HCWs and frequent visitors, whose circumstances may differ significantly from the perspective explored in this work. Another notable limitation of the study design was the size and (restricted) diversity of participants. We had 19 participants with all-female screening staff due to the limited number of male nurses in the children’s hospital. This could impact our findings, as we did not have an opportunity to explore any gendered distinctions we suspect may have occurred (particularly in relation to perceptions of authority and power). Finally, the eGate system was conceptualised, tested, and deployed in a very complex environment during the COVID-19 pandemic (during its design process, the environment was rapidly changing), and therefore access to participants and stakeholders was particularly challenging. The environment and COVID-19 response in Sydney, Australia might have influenced the perception of the eGate system. It should also be noted that at the time of running the study, the pandemic and the world’s response (i.e., restrictions, vaccines, and testing) was unfolding. The gold-standard measures for responding to COVID-19 today are different to what they were then (which was pre-rapid antigen tests, widespread vaccination, etc.) (COVID-19: a chronology of state and territory government announcements—https://www.aph.gov.au/About_Parliament/Parliamentary_departments/Parliamentary_Library/pubs/rp/rp2021/Chronologies/COVID-19StateTerritoryGovernmentAnnouncements (accessed on 16 September 2021)). Our eGate may evolve to adopt future iterations that include advances in improving the range, types, and validity of screening tests [44].

## 6. Conclusions

Digital health-screening measures are continuing to evolve in response to the COVID-19 pandemic in order to keep everyone safe, particularly those in hospitals which contain vulnerable people and those working to treat them. In this paper we reported on the experiences of concierge screening staff who were stationed alongside the eGate, a digital COVID-19 health-screening system eventually deployed at three main entry points at a large children’s hospital. Using qualitative data collected from screening staff through in situ interviews and questionnaire responses, we performed a thematic analysis which revealed key socio–technical themes informing the deployment of digital health-screening systems in hospitals. We then provide six key design recommendations which point towards screening systems which balance system authority with the staff who work alongside them, can easily be understood and engaged with from a wide audience, and are reactive to individuals using them and to possible technical failures.

## Figures and Tables

**Figure 1 ijerph-20-03899-f001:**
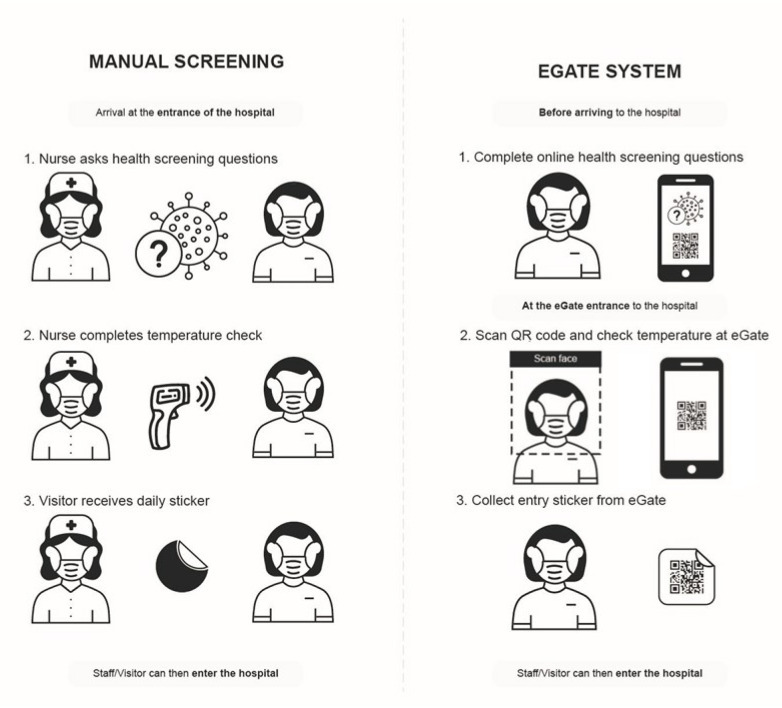
Change in agency from manual to eGate. **Left**: Manual screening methods; **Right**: eGate digital screening system.

**Figure 2 ijerph-20-03899-f002:**
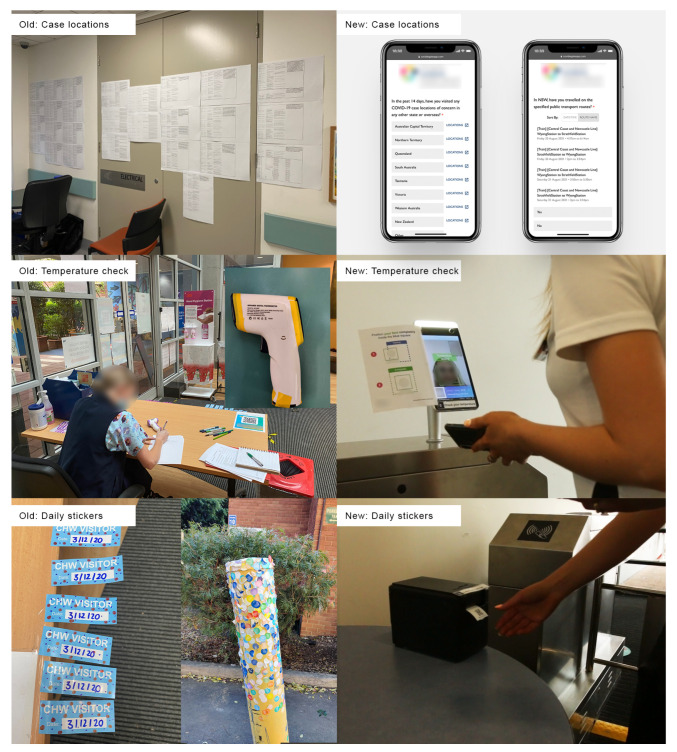
Contrasting the manual system with the eGate system.

**Table 1 ijerph-20-03899-t001:** Information for each participant: ID, age-group, role (assistant in nursing (AIN), undergraduate assistant in nursing (UG AIN) registered nurse (RN), enrolled nurse (EN), and the study group they were involved in.

	Age Group	Role	Study Group
P1	18–25	UG AIN	Interview
P2	18–25	UG AIN	Interview
P3	18–25	AIN	Interview
P4	26–30	AIN	Interview
P5	31–40	AIN	Interview
P6	18–25	AIN	Online survey
P7	18–25	AIN	Online survey
P8	18–25	AIN	Online survey
P9	18–25	UG AIN	Online survey
P10	18–25	UG AIN	Online survey
P11	18–25	UG AIN	Online survey
P12	18–25	UG AIN	Online survey
P13	26–30	EN	Online survey
P14	26–30	EN	Online survey
P15	31–40	EN	Online survey
P16	31–40	Nurse manager	Online survey
P17	40+	RN	Online survey
P18	40+	Pre-admission nurse	Online survey
P19	40+	Clinical nurse Specialist	Online survey

## Data Availability

The data that support the findings of this study are available on request from the corresponding author, AP Wang. The data are not publicly available due to restrictions e.g., their containing information that could compromise the privacy of research participants. The study’s pre-registration can be found at https://osf.io/tpxur (accessed on 22 December 2020).

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
