# Peer review of "Designing Digital COVID-19 Screening: Insights and Deliberations"

_ijerph, 2023, doi:10.3390/ijerph20053899_

Round 1
Reviewer 1 Report
My main objection to this article is that it is a description of a specific case and lacks comparative material to be considered fully original. In addition, some of the solutions proposed by the authors of the article, such as those involving aconcealed button button, resemble solutions known from totalitarian systems or those implemented at airports to detect potential terrorists. It seems that the goal the authors have set for themselves does not explain such measures. In a democratic society, all actions of all services, especially inspection services, should be official.
Reviewer 2 Report
Results should be presented in more clear form
Reviewer 3 Report
There have already been many studies regarding the application of Digital Technologies in Health Care During COVID-19. User-friendliness of a healthcare provider perspective has been proven.
A qualitative research approach is necessary, but it is time for new interventions or improvement in the direction of increasing the validation of screening tests.
Round 2
Reviewer 3 Report
All the suggestions presented in the first review are reflected.